



# Virtual field trips as a tool for indirect geomorphological experience: A case study from the southeast part of the Gulf of Corinth, Greece

Niki Evelpidou[1], Anna Karkani[1], Giannis Saitis[1], Evangelos Spyrou[1]

[1]Faculty of Geology and Geoenvironment, National and Kapodistrian University of Athens, Panepistimioupoli, 15784, Greece

*Correspondence to*: Niki Evelpidou (evelpidou@geol.uoa.gr)

**Abstract.** Field trips are an essential part for geoscience students, as the field is intrinsic for understanding what they are taught in the classroom. Yet, distance learning has never been more necessary than today. Despite their significance in the students' education, field trips cannot be performed under the present conditions with the COVID-19 pandemic. Educators are called to find, use and evolve various tools in order to offer students quality education, with an effort to eliminate the drawbacks of the

lack of physical contact and "live" field work. Virtual field trips are one such tool, through which one can virtually see any place in the globe by means of a computer, tablet, or even mobile phone, without physically visiting it. In this paper, we present the results of a virtual field trip developed for students following Geomorphology course of the Faculty of Geology and Geo-environment, National and Kapodistrian University of Athens; it can, however, be used from other universities with similar courses, not only in Greece, but in other countries as well. The purpose of this study is the evaluation of virtual field trips, both

as an alternative and/or substitute of *in situ* field work and as a means of preparation for "live" field trips, by taking into consideration the students' views, through an anonymous questionnaire. Our findings suggest that virtual fieldtrips are useful for geoscience students, and they provide a good alternative during restriction periods, and although they can under no circumstances substitute real field trips, they can be a valuable additional tool when preparing for a live field trip.

## 1 Introduction

Field trips are an essential part of a geologist's education. They are very helpful in understanding the geological processes that have taken place in a study area (e.g. Hurst, 1997). They offer both trainees and educators the ability to communicate to each other, co-operate and develop a team spirit (e.g. Clark, 1996). In the field, one can recognize several geological structures and landforms and comprehend the processes that have led to their formation (e.g. Hurst, 1997).

Virtual field trips are an alternative way to study an area. It is a tool via which one can virtually see any point of interest, rather

than physically visiting it (e.g. Stainfield et al., 2000; Carmichael & Tscholl, 2011). It is much more than simply presenting images and slides. One can view photos, videos, as well as satellite images, and from different aspects, both in two and three dimensions. Additionally, in this way, literally any place in the globe can be visited, whereas their size is theoretically unlimited, and they can safely be stored in any device. They are accessible from almost any place, meaning that people from



different countries can attend them simultaneously (Stainfield et al., 2000), and they are more necessary than ever before,
given the COVID-19 pandemic conditions.

It is an easy way to both see and learn about any place of interest and understand the principles of geology, tectonics, geomorphology and so on, and explain them to a student or pupil, i.e. to use as a teaching method. Even if it is very difficult or impossible for one to visit a place in person, one can see almost every point of interest via certain computer programs, such as ArcGIS and Google Earth, as if they were actually there. In addition, this tool is very easy to use, whereas it is accessible
from any device with an internet connection, such a personal computer, a tablet or a mobile phone (Çalışkan, 2011); it may therefore be used in the classroom, from their office and even from home.

Additionally, virtual field trips are also useful when it comes to health aspects or situations where it is forbidden to visit a place. They are particularly useful when the said place is very far away and/or not accessible (Hurst, 1997). For instance, it is very difficult for students of the Mediterranean countries to visit countries in North Europe, not to mention Africa or America.
Via this tool, however, they can "visit" these countries without actually being there.

Virtual field trips are also useful when certain sites to be studied cannot be visited, for safety, time or weather reasons (Çalışkan, 2011). Virtual field trips are interactive and give students the opportunity to explore sites of interest themselves (e.g. Stainfield et al., 2000), as well as present their own work, if related to these sites (Gratton, 1999), as opposed, for instance, to a mere power point presentation, where one does not have this ability.

Even though they do not substitute the actual field work, they offer a valuable indirect field experience, including the chance to see places, landmarks and geological formations that are not easily accessible, to understand the geological evolution of a study area, to construct geological and geomorphological maps and much more. Furthermore, virtual field trips can contain links of other auxiliary material, such as papers, maps or field guides, thus giving students the ability to further study an area or site of interest (e.g. Stainfield et al., 2000). And what is more, not only are they very helpful when it comes to performing
a field trip from a distance, but they can be of paramount significance when planning an actual field trip as well. Upon study of an area's geological and geomorphological evolution, this tool can be used in order to prepare which sites are to be visited for sample collection, measurements of tectonical and stratigraphical characteristics, observation of specific landforms and so on (Gilmour, 1997; Stainfield et al., 2000; Cliffe, 2017), and focus on more practical issues in the field.

Every year, the Faculty of Geology and Geoenvironment of the National and Kapodistrian University of Athens (NKUA)
organizes one field trip for every main course. These field trips are mandatory and free for the students and take place in various sites of Greece. For the course of Geomorphology, the field trip normally takes place in the Corinthian Gulf. However, the restrictions due to the COVID-19 pandemic did not allow its actualization. To address this issue, we created a virtual field trip in the same sites that would normally be visited, through the Google Earth platform, in order to aid the students to understand the principles of the course.

This paper focuses on the said virtual field trip. Initially developed as an alternative of the actual field trip that was thwarted due to the COVID-19 pandemic, our goal is to showcase the significance of virtual field trips via this "excursion". The main purpose of this work is to evaluate virtual field trips and to what extent they can be used as an alternative to "real" field work,





that is to what extent they can aid students to comprehend the fundamentals of a course or discipline, especially in situations where visiting an area is not feasible.

**2 Materials and Methods**

The virtual fieldtrip addresses topics such as coastal geomorphology, Holocene relative sea level changes and sea-level indicators, palaeogeographic evolution, morphotectonics and palaeo-earthquakes. Initially, bibliographical references were studied on the geological, geomorphological, tectonical and archaeological characteristics of the study area. Additionally, topographical, geological and other maps, as well as satellite images were studied. The individual stops of the field trip were

visited and documented through aerial photography, photographs and videos.

For the complete overview of the field trip, a field trip guide was authored, providing information about the wider area and each of the stops individually. The collected data were imported into ArcGIS software, for data analysis and the development of thematic maps. The visualization of this virtual field trip was accomplished through the web platform of Google Earth, through which it is possible to virtually visit and study the landforms of the area, as well as their relationship to the ancient

anthropogenic interventions. The platform was further enriched with photographic and video material from field work and educational field trips in the area. The presentation of the virtual field trip was accomplished in 134 third-year students of the Faculty of Geology & Geo-environment. Upon its completion, an anonymous questionnaire was distributed to the students in order to evaluate the virtual field trip in terms of their expectations and its usefulness.

**3 Study area**

The Gulf of Corinth is among the most tectonically active regions of Greece (e.g. Moretti et al., 2003; Gaki-Papanastassiou et al., 2007; Jolivet et al., 2013; Charalampakis et al., 2014; Lazos et al., 2020). It lies in the central part of Greece and segregates the Peloponnese, to the south, and Central Greece, to the north. It has a total length of 120 kilometers and a width of more than 27 kilometers, whereas its maximum depth reaches 870 meters (e.g. Gaki-Papanastassiou et al., 2007; Charalampakis et al., 2014). Its direction is WNW-ESE, while at its eastern edge, it is separated by Perachora peninsula into two lesser gulfs, i.e.

the Gulf of Alkyonides, to the north, and the Gulf of Lechaeum, to the south (Fig. 1).

The broader area is characterized by intense seismicity and tectonic movements. The gulf itself is a graben, yet the mainland is characterized by many individual horsts and grabens. Along its vast scientific significance apropos structural geology and stratigraphy, there are several areas with a vast interest regarding geomorphology, tectonics, relative sea level changes as well as geoarchaeology (e.g. Gaki-Papanastassiou et al., 2007; Koukouvelas et al., 2017; Gawthorpe et al., 2018; Zhong et al., 2018;

de Gelder et al., 2019; Emmanouilidis et al., 2020), making the area ideal for students to understand coastal geomorphological processes in a tectonically active area.



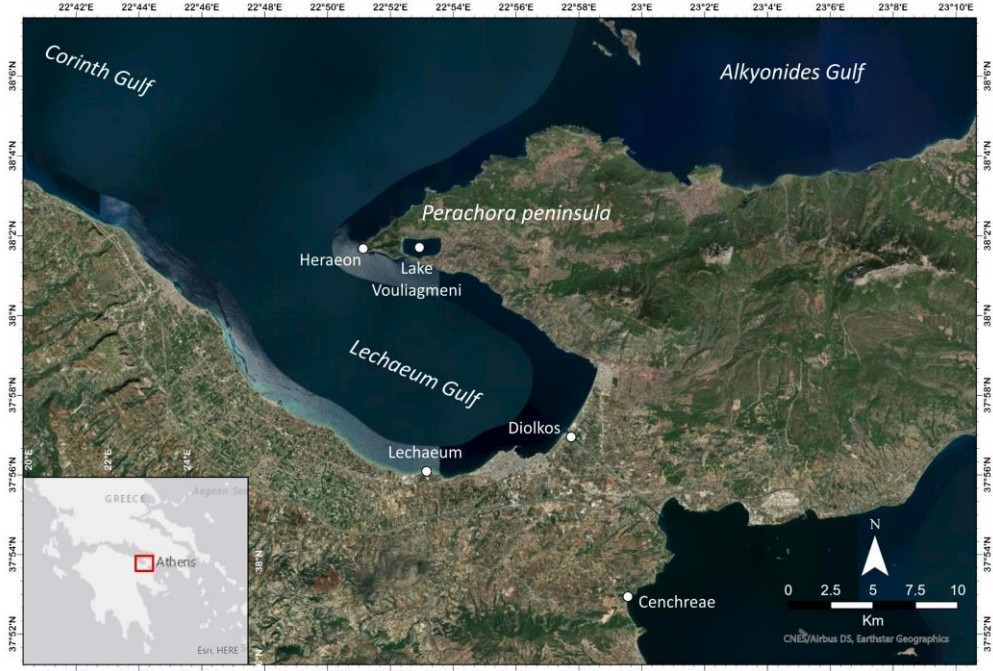

**Figure 1: Location of the sites discussed in the text. Inset map shows the location of the study area (red box). (Created using ArcGIS Pro).**

The designed virtual field trip can be found here and consists of five stops: Lechaeum, Cenchreae, Diolkos, Lake Vouliagmeni

and Heraeon. For each stop, the web platform contains geomorphological and geoarchaeological information, as well as its

evolution from Antiquity to the present day, concerning both the tectonic conditions and its ancient history (Fig. 2).

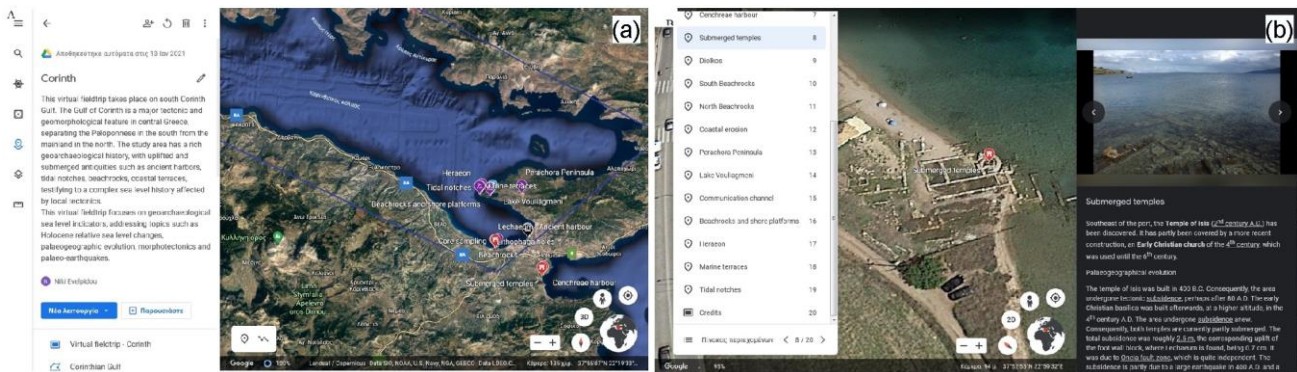

**Figure 2: (a, b) Examples of the virtual fieldtrip developed on the Google Earth platform. © Google Earth 2021**





## 3.1. Lechaeum

Ancient Corinth had two harbours: Lechaeum on the Corinthian Gulf and Cenchreae on the Saronic Gulf. Lechaeum (Lechaion or Lecheae) was an artificial harbour of the Archaic Period and is estimated to have been constructed in the 6th century B.C. (Rothaus, 1995; Stiros et al., 1996). The port is characterized by a peculiar helical geometry, resembling a natural channel (Morhange et al., 2012). In the area of Lechaeum, beachrocks can be observed, partly covering a construction (probably warehouses of the ancient port), dating back to the Classical Period (490-323 B.C.), meaning that it formed before 2400 B.P. The fact that the beachrocks have covered the ancient construction probably indicates a subsidence of the area, followed by the formation of the beachrocks, and eventually the uplift of the area (Stiros et al., 1996; Morhange et al., 2012).

Stiros et al. (1996) used uplifted marine bivalves in growth position in order to date the co-seismic event of the uplift (Fig. 3). The biological indicators suggest an uplift of at least 70 cm, which took place between 500 and 200 B.C., probably around 340 B.C. Upon radiometric dating, the said uplift took place around $2470 \pm 45$ B.P. ($375 \pm 120$ cal. B.C.) (Morhange et al., 2012). Additionally, two boreholes took place in the currently dried basin (Fig. 3) in order to analyze the coastal stratigraphy and understand the evolution of the area (Morhange et al., 2012). The first core reached a depth of 265 cm and was characterized by the absence of sediments from the Archaic Period, which was justified by an intermittent deepening of the harbour (Morhange et al., 2012). According to Marriner & Morhange (2007), that is actually the case for most Mediterranean ancient harbours.

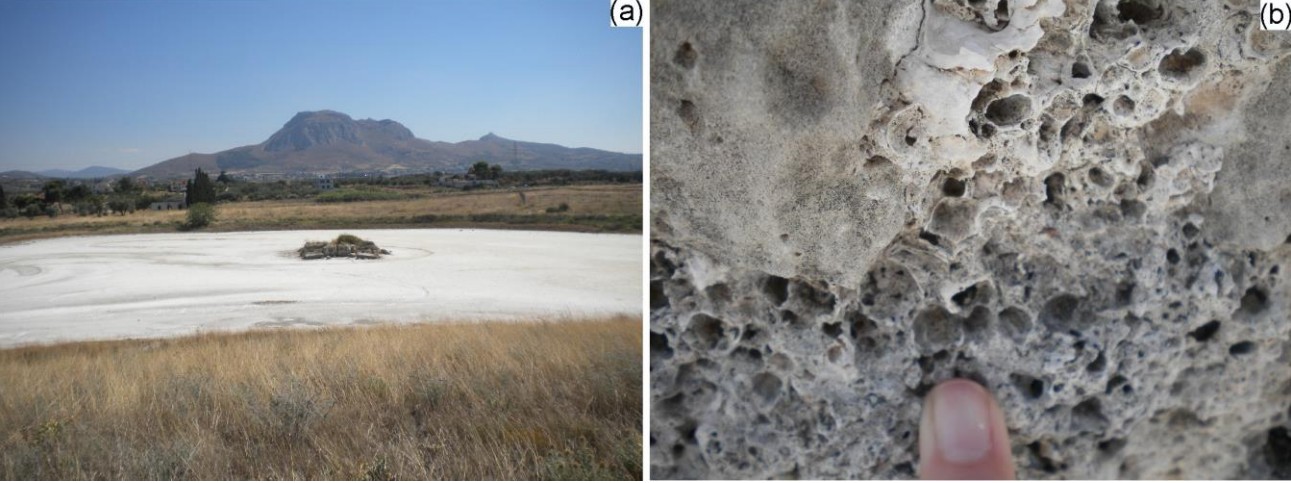

**Figure 3: (a) Part of the main silted harbor in Lechaeum, in the middle of which a rectangular structure is visible. Corings in this area revealed that tectonic uplift in combination with the location of the harbor in serpentine depression were the primary factors for its abandonment (Morhange et al., 2012). (b) Closer view of the structure where uplifted *Balanus perforatus* fossils in growth position suggest a biological sea level at +1.2 m, with the age of the uplift at $375 \pm 120$ cal. B.C. (Morhange et al., 2012).**

This site was chosen so that students could comprehend the geomorphic processes that have affected the area, both marine, i.e. changes in the sea-level over the centuries, and terrestrial ones, i.e. the increased sediment yield due to both erosion processes





and human interventions, which resulted in the deepening of the ancient harbour. In other words, the students would be able to comprehend its palaeogeographical evolution, as well as the relationship between tectonics and sea-level changes and the contribution of geoarchaeological indicators and core sampling for the palaeogeographical reconstruction of a site.

### 3.2. Cenchreae

Cenchreae (Kenchreai or Kechries) was the eastern harbour of ancient Corinth, located northeast of the Isthmus of Corinth. It was a natural harbour, as opposed to Lechaeum. Cenchreae is surrounded by three main seismic sources, namely Cenchreae, Loutraki and Aghios Vasileios faults, which have a potential of causing a subsidence of the area, as it is located on the hanging-wall block (Papanikolaou & Roberts, 2011).

Southeast of the harbour, the temple of Isis has been discovered, built in 400 B.C. It has partly been covered by a more recent

construction, an early Christian church of the 4th century. The temple of Isis is located at a lower altitude than the early Christian basilica, whereas the two constructions are currently located below sea-level. One can thus understand the palaeogeographical evolution of the area. The temple of Isis was built, consequently the area was submerged, and the early Christian basilica was built at a higher altitude. The fact that the early Christian basilica is submerged as well indicates that the area has undergone subsidence anew. The total subsidence of the area is roughly 2.5 m, while the corresponding uplift of the foot wall block,

where Lechaeum is found, is 0.7 cm. The subsidence of Cenchreae is owed to Oneia fault zone (Maroukian et al., 1994).

Rothauss (1996) partly attributed the subsidence of the area to a large earthquake in 400 A.D. and partly to a subsequent earthquake, arguably in 600 A.D. According to Rothaus et al. (2008), the subsidence of Cenchreae took place after 80 A.D. According to Papanikolaou & Roberts (2011) and Noller et al. (1997), at least four earthquakes in the broader area contributed to the subsidence of Cenchreae harbour (400 A.D., 1756, 1858, 1928).

This site is a characteristic example of an area where tectonic control is intense. Though limited in surface, one can observe both tectonic subsidence and uplift. The reason why this site was chosen is for the students to understand how the said tectonic movements can not only affect but determine the geomorphological evolution of an area. Moreover, prominence could be given to the significance of archaeological information in the palaeogeographical evolution of an area.

### 3.3. Diolkos

Diolkos was a paved road used for transporting ships from the Corinthian to the Saronic Gulf, i.e. between Lechaeum and Cenchreae harbour, and vice versa, via a wheeled vehicle, where the ships were moored, once exempt from their cargo (Fig. 4a). It was constructed in the 6th century B.C. Its width was 3.5 to 6 m and its length reached 8 km (Koutsouba & Nakas, 2009).

**Figure 4: (a) A panoramic view of the western part of the Isthmus. Diolkos can be seen on the right. (b) The younger beachrock is visible covering part of Diolkos paved road. According to Pirazzoli et al. (2012 two different subsidence phases and an intermediate uplift can be distinguished. (c) The area of Diolkos suffers from intense coastal erosion, and a large part of Diolkos has been destroyed due to erosion, while the coast is under a continuous coastal retreat mainly owed to the waves generated by large ships.**


The broader area is characterized by the presence of beachrock formations. In Poseidonia, at the exit of the Gulf's channel towards the Peloponnese, Diolkos remnants overlie a coarse beachrock. Beachrock fragments were used for the paving of the





western part of Diolkos, meaning that the formation, or at least its last phase, is newer than the construction of Diolkos. In the same part of Diolkos, a fine beachrock, dipping to the northeast, covers the pavement and the older, coarse beachrock.

According to Pirazzoli et al. (2012), in Diolkos, as well as in the western part of Lechaeum, two different subsidence phases and an intermediate uplift can be distinguished. Initially, the paved road of Diolkos was incorporated into the older, coarse beachrock and at a later time, it subsided during one or more tectonic events. The newer beachrock was a result of the cementation of coastal sediments which accumulated on the surface of Diolkos, after an event of tectonic uplift (Fig. 4b). During the following subsidence phase, part of the newer beachrock formation was submerged.

The area is also characterized by intense coastal erosion and coastline retreat. A large part of Diolkos has been destroyed due to erosion and coastline retreat, which in turn is owed to the waves generated by large ships. Up until 30 years ago, in the northern part of the Isthmus and towards the Gulf of Corinth, another part of Diolkos existed, which today is wholly destroyed. After the opening and widening of the Isthmus, the balance in the coastal zone was disrupted. Additionally, the excavation of Diolkos, as well as the removal of large quantities of material, resulted in coastal retreat (Fig. 4c). The transversal of large

ships and the consequent waves have intensified the coastal erosion.

In this site, students can be familiarized with coastal morphodynamic processes and the recognition of previous coastal events, owed to changes in the sea-level and human interventions, as reflected in coastal features such as beachrocks. Issues of coastal erosion and its documentation are also addressed in this stop, discussing methods and techniques to monitor the phenomenon as well as possible protection measures. Furthermore, the students can deepen their knowledge on relative dating methods,

combining archaeology and geomorphology.

### 3.4. Lake Vouliagmeni

Lake Vouliagmeni (or Lake Heraeon) is a lagoon in Perachora Peninsula (Fig. 5), formed as a karst cavity and is separated from the sea via an isthmus with a minimum width of 8-10 m. The most important morphodynamic characteristic is a double

escarpment, in an E-W direction, created by Pisia-Schinos normal fault zone, which is responsible for the extension of this part of the Corinthian Gulf by 0.45 mm per year (Maniatis, 2006). The most representative seismic events owed to this fault zone were those of 1981. Three strong earthquakes hit the area and many superficial ruptures were observed (Jackson et al., 1982, King et al., 1985). The average seismic displacement was 0.5-0.7 m (Collier et al., 1992). The earthquakes caused both uplift and subsidence in the broader area (Vita-Finzi & King, 1985; Hubert et al., 1996). Morewood & Roberts (1999) have

noted uplifted marine terraces along the south coasts of the peninsula (between Loutraki and Lake Vouliagmeni).

According to Gaki-Papanastassiou et al. (1997), in the broader area, beachrocks have been found at a maximum altitude of +3 m, forming four levels (+3, +2, +1 and +0.4 m). The ones at +1 and +0.4 m are wider and cover a significant part of the coastal zone, whereas the ones at +3 and +2 m, are only found locally. It is worth mentioning that along the lake's coast, lacustrine beachrocks have been found at an altitude of 1.4-1.6 m, underlying the Early Helladic settlement, which indicates that these

beachrocks are older than the settlement (>3200 B.C.) (Gaki-Papanastassiou et al., 1997). Stiros & Pirazzoli (1998) have also



noted the foresaid beachrocks, noting two platforms in the coast near the lake and at an altitude of +0.9±0.2 m and +0.4±0.2 m, consisting of cemented coastal material. These platforms are possibly related to the uplifted coastline (at +1.1 m) near Heraeon. They continue for at least 5 km, with small differences in altitude, which implies that the coast corresponds to a rigid tectonic block, which has been uplifting and subsiding during at least the last 6,000 years, with no significant deformation

(Pirazzoli, 2012).

One of the most intriguing landforms of this site are the lacustrine beachrocks, which are very rare features. Their observation can aid the students understand their formation processes better. During the "live" field trip, they would have the ability to view, study and create geomorphological maps as well. The geomorphological mapping is a very important aspect of the field work. Yet, it could not be performed virtually.

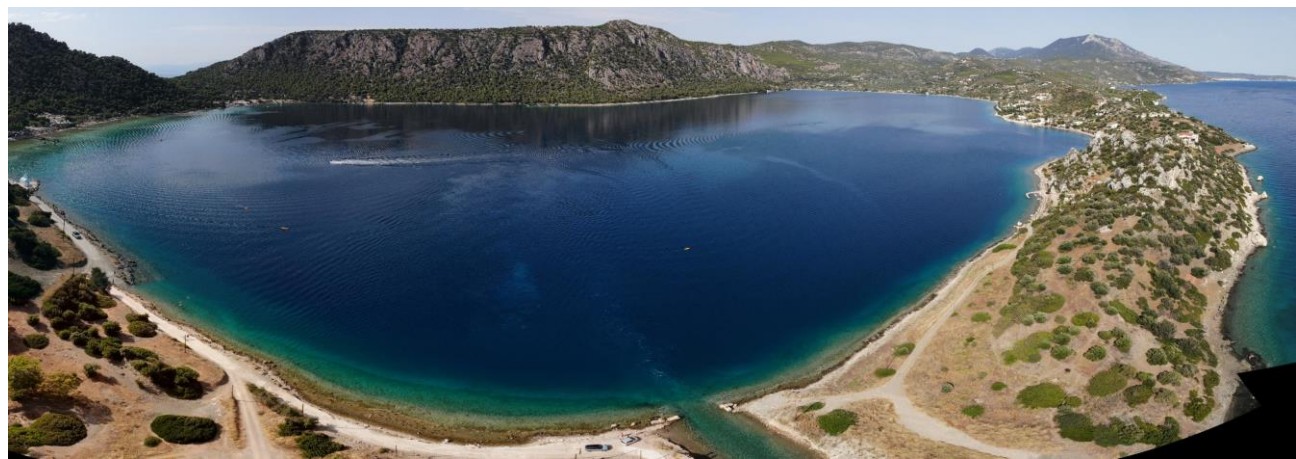

**Figure 5: (a) Aerial view of Lake Vouliagmeni, formed as a karst cavity. On the bottom part of the image, the artificial channel is visible, constructed a century ago, connecting the lake with the sea. According to Stiros (1995), the level of the lake corresponded to the sea level, even in antiquity.**


## 3.5. Heraeon

The small harbour of Heraeon is located at the far west edge of Perachora Peninsula, to the west of Lake Vouliagmeni, where remnants of the temple of Hera have been found. The temple is believed to have been constructed in the 8th century B.C. The

area has suffered many earthquakes (Payne, 1940; Blackman, 1966), one of which took place in the end of the 8th century B.C. and the temple was moved westwards. An earthquake, which took place in the 2nd century A.D., destroyed the temple (Gaki-Papanastassiou et al., 2007).

In Perachora peninsula, there are at least three uplifted Pleistocene marine terraces at 6.5-13 m, 25-28 m and 45 m (Fig. 6a). Several authors (e.g.Vita-Finzi, 1993; Pirazzoli et al., 1994; Dia et al., 1997, Leeder et al., 2005) have stated that the peninsula

has undergone constant uplift from Late Quaternary to Holocene, through radiometric dating of marine fossils found within uplifted marine deposits. These terraces indicate that the mean uplift rate of the area is 0.2 mm/year for the last 220,000 years.



Additionally, in the area, remnants of uplifted tidal notches have been found in Mesozoic carbonate cliffs, at altitudes of +3.2±0.2, +2.6±0.2, +1.7±0.2 and +1.1±0.2 m (Pirazzoli et al., 1994) (Fig. 6b, c). A recent study by Schneiderwind et al. (2016) revealed four palaeo-shorelines at + 0.4 m, + 1.9 m, + 2.1 m and + 2.3 m ± 0.2 m. Schneiderwind et al. (2016) used

high resolution laser scanning, which offers the ability to study landforms in detail, as well as tracing multiple notches, which represent palaeo-shorelines during successive tectonic uplifts.

In this site, students are better acquainted with the morphotectonic evolution of coastal areas, based on sea-level indicators, both geomorphological, such as marine notches and terraces, and biological. In addition, emphasis is given to various dating methods, and consequently to the rates of relative sea-level change.

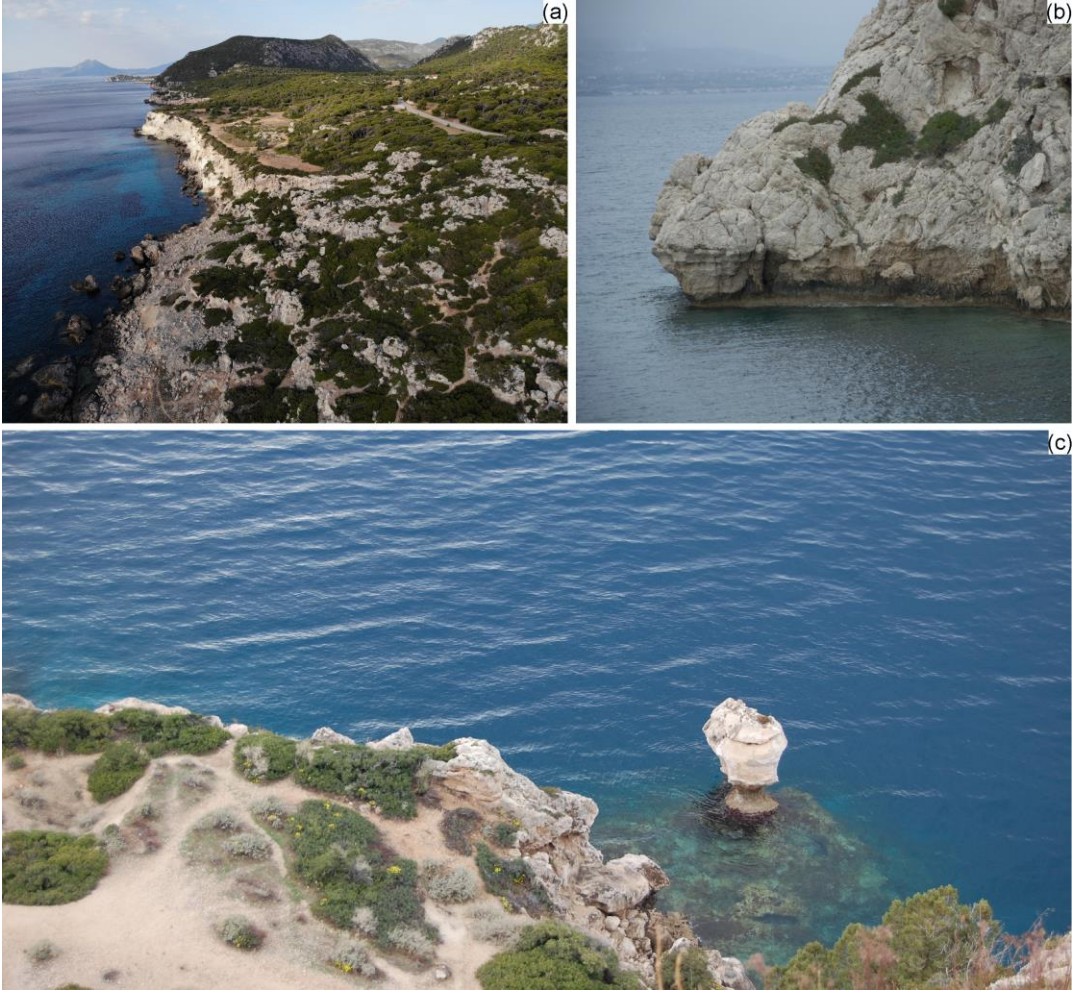

**Figure 6: Morphotectonic indicators in Perachora peninsula, revealing an uplifting regime in the area from Late Quaternary to Holocene. in the form of (a) Pleistocene marine terrace at about 20-25 m, which was dated at about 128±3 ka, based on a sample of Cladochora caespitosa (Vita-Finzi, 1993). (b) Multiple remains of uplifted tidal notches were studied by Pirazzoli et al. (1994) at altitudes of +3.2±0.2, +2.6±0.2, +1.7±0.2 and +1.1±0.2 m, documenting uplift events since 440-4320 B.C. (c) Characteristic mushroom shaped rock, with an uplifted tidal notch, about 150 meters to the east of (b) (Pirazzoli et al. 1994).**




## 4 Results

The anonymous questionnaire was filled by 51 students from the Faculty of Geology and Geo-environment (NKUA), who follow the course of Geomorphology. About 82% of the students who attended the virtual field trip had not attended another virtual field trip before. 80% of them would attend one such field trip again, even when the restrictions due to the pandemic were abrogated, both for educational reasons and in order to prepare themselves for an actual field trip. The main reasons were the ability to "visit" remote areas and view satellite images, the ease of their usage, the saving of time and the ability to panoramically view a site. 78% stated that they would prefer to use virtual field trips instead of a PowerPoint presentation, for instance, in order to educate other trainees.

Approximately 90% of the students stated that it offered them useful information about the studied sites. About 80% of the students were aided in the understanding of the fundamentals of the course of Geomorphology. Three quarters (75%) declared that the means of this field trip was satisfactory, given the pandemic conditions.

However, more than 88% of the students stated that a virtual field trip cannot substitute an actual one. Amongst these students 49% mentioned the restrictions of virtual field trips in comparison to actual field work. For more than half (56%), the most important drawback of virtual field work is the incapability to observe geological structures and landforms with their own eyes. According to them, through virtual field trips, one cannot escalate their observation skills as a geologist or geology student, whereas one cannot observe a site from different viewpoints, as opposed to live field work. Additionally, through an online platform, one cannot view and/or recognize a landform or feature in detail, especially one that is relatively small in size. Amongst the students who considered that a virtual field trip cannot substitute an actual one, for 32% another restriction of virtual field trips, was the decreased interaction with other students, educators, and nature. Through virtual field work, they can neither collect rock and other samples nor feel or touch the rocks and other geological features.

## 5 Discussion

The purpose of this virtual field trip was for geoscience students to understand the geological and geomorphological processes that have formed the relief of the studied sites, both endogenous, i.e. tectonic uplift and subsidence, and exogenous, i.e. erosion, deposition, biological activity etc. In its entirety, this virtual field trip aimed to aid trainees in recognizing several coastal landforms, understanding their formation processes, as well as identifying several geoarchaeological sea-level indicators and understanding the recent evolution of the individual studied sites based on their observations. In addition, we intended to help our students understand the most common geochronological methods and their applications.

What is more, the usage of the Google Earth platform could give the students the ability to view the sites of interest in three dimensions and from different aspects, thus understanding their spatial arrangement. Moreover, this virtual field trip aimed to their familiarization with geological and geomorphological maps, as well as satellite images.

According to the students' answers to the questionnaire and their comments, they were aided by this virtual field trip and the provided multimedia material. Most students obtained useful information about the virtually visited sites and they understood



better the principles of the course of Geomorphology, as well as other geological courses. What is more, most of the students would attend and/or create virtual field trips, even after the restrictions due to the COVID-19 pandemic have been abrogated,

not only in order to prepare themselves for *in situ* educational field trips or field research, but as a means of education as well. Virtual field trips are not, or should not be, used to replace the actual field work (e.g. Gilmour, 1997). However, given that they offer the students interactivity in comparison to other educational methods that can be used in the classroom (e.g. Ramasundaram *et al.*, 2005; Cliffe, 2017), they can be used as a means of preparation for actual field work, concerning both the sites to be visited and the students' obligations when they visit them. Moreover, virtual field trips are very useful in visiting

remote areas and/or sites that are not accessible (Stainfield *et al.*, 2000) and they are a valuable tool for disabled students, who are unable to visit many areas (Hurst, 1997; Gilley *et al.*, 2015). What is more, existing field work can be improved by means of a virtual field trip, as the same sites can easily be "visited" anew, and more observations can be made. Existing field work can be improved with better before-hand preparation, for panoramic observations and for revisiting complex sites/locations.

Yet, as helpful and useful as they can be, virtual field trips do have certain drawbacks in comparison to actual field trips that

are worth mentioning. These drawbacks do not concern solely academic students, but all education levels, such as pupils (e.g. Spicer & Stratford, 2011; Mead *et al.*, 2019). Initially, educational field trips as well as scientific field work pose a unique experience to both trainees and educators, as they have the ability to communicate to each other in the flesh, to co-operate and develop a team spirit (e.g. Clark, 1996). Additionally, understanding the fundamentals of any course in the field is highly aided by the contribution of the students' senses. The latter do not refer solely to seeing and observing landforms, but include sound,

smell, touch, even taste. Virtual trips deprive the students of these types of interaction with each other, with the educators, with nature and the landscape (e.g. Çalışkan, 2011; Han, 2020).

What is more, for the vast majority of students and especially geoscience ones, actual field trips are far more beneficial in recognizing geological structures and landforms and understanding their formation processes and their significance in geological surveys (e.g. Hurst, 1997). Within a few hours in the field, one can learn and comprehend much more than during

a significant amount of time in the classroom. And it is to be mentioned that what is taught in the field is far more likely to be remembered than in the classroom. For instance, it has already been mentioned that one of the purposes of this field trip was to aid students to view and use geomorphological maps and create their own ones. Yet, this could not be performed through the virtual version of the trip, as high accuracy aerial photos and stereoscopical observations would be needed. Even though this can be improved in the near future, small-scale landforms, stratigraphic boundaries and other features could not be

observed well, as their scale is not large enough to be distinguished with the used platform. However, other means for the design of virtual fieldtrips, such as 360 videos have the possibility to improve certain aspects of such an activity, offering a more detailed view on particular geomorphological features.





# 6 Conclusions

Virtual field trips can supplement the physical fieldwork and are a succor in any field trip and/or field work, offering both
students and educators/researchers the ability to prepare themselves. According to the student responses, the designed virtual
fieldtrip achieved its goal, as they improved their understanding on the course of Geomorphology, and they were satisfied by
attending a virtual fieldtrip, given the pandemic conditions. Virtual field trips can be a useful tool for additional field work in
places with difficult access, for people with mobility problems, for improving existing field work with better before-hand
preparation, for panoramic observations and for revisiting complex sites/locations.

Yet, under no circumstances can they substitute actual field trips. The students' interaction with each other, the educators and
nature are essential in the effectiveness of a field trip and is only the case when this trip is physical. It is also worth noting that
the feedback of geoscience students is essential to improve virtual field trips and create more-inclusive activities.

## Author contribution

NE conceived and directed the project. NE, ES, GS and AK contributed to the design of the virtual fieldtrip. NE, AK, GS and
ES contributed to the analysis of the results and the writing of the paper.

## Competing interests

The authors declare that they have no conflict of interest.

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
