# Peer review of "Virtual field trips as a tool for indirect geomorphological experience: A case study from the southeast part of the Gulf of Corinth, Greece"

_Geoscience Communication, 2021_

## Referee Comment (RC2)

[referee-annotated manuscript omitted]

---

## Author Comment (AC1)

**Would you attend again a virtual fieldtrip?**

[Figure]

Yes ■ No ■

**Why would you attend a virtual fieldtrip again?**

[Figure]

- ■ Reduced cost
- ■ Easy to use
- ■ Time saving
- ■ Reduction of physical fatigue
- ■ Possibility of additional educational excursions
- ■ Possibility of visiting remote areas
- ■ Panoramic view of areas
- ■ Three dimensional view

**Would you use virtual fieldtrips instead of a powerpoint to educate to teach on a particular topic?**

[Figure]

Yes ■ No ■

**Do you think virtual fieldtrips can substitute real fieldtrips?**

[Figure]

Yes ■ No ■

---

## Author Response (AR1)

**Response to reviewers**

We would like to thank the editor and the reviewers for their comments, which helped to improve our manuscript. We have revised the manuscript accordingly, and we provide details here. Our responses are provided in blue.

**Referee #1**

The manuscript entitled "Virtual field trips as a tool for indirect geomorphological experience: A case study from the southeast part of the Gulf of Corinth, Greece" is of a very interesting topic. It presents a virtual fieldtrip with five stops at the southeastern part of the tectonically active Gulf of Corinth created using the Google Earth platform. The paper describes briefly the geomorphology – geoarchaeology of the sites and presents the opinion of the students about their experience of visiting the area virtually.

It is very important that the authors clarify that virtual fieldtrips cannot and should not replace real fieldtrips but could act as a tool to better understand the geology-geomorphology of an area or for a student to "remember" what he has seen in the field. Additionally, activities such as "real" sampling (of any kind of samples and for any purpose) and mapping of landforms or studying the stratigraphy of a profile are necessary during a fieldtrip.

The manuscript is well written and well structured. Since it is a "different type of paper", I mean not a case study of a "classic" geomorphological research, I cannot judge if the description of the sites should be given in that detail since the results of the paper include only the answers of the students. I am not a native speaker but I think that the paper reads well.

Thank you for the comments. Regarding the sites' description, we have reduced details and provide only the basic information and the justification of why each site was included.

Here are some very minor suggestions that in my opinion could improve the final version of the paper:

At the "Materials and Methods" section the authors should add a couple of sentences about the technical part of the design of the platform in Google Earth.

Thank you for the suggestion. We provide a few more details regarding the design of the virtual fieldtrip in Google Earth platform (see lines 75-80 of the revised manuscript).

The results regarding the opinion of the students are well presented and explained. The findings from the answers of the students could be presented in diagrams. Maybe just one figure showing the percentages of the student's answers in the main questions could be added.

Thank you for the suggestion. We have now added a new figure (Fig. 7), which presents students responses in some characteristic questions.

**Referee #2**

Overall, the concept is a really good one and I applaud the authors for trying this technique and monitoring its impact on their students. I would have liked to see much more about how the students were evaluated and the results from the data that was collected. I also felt that there was too much detail on each of the stops and not enough about the interface and the way that the students engaged with it. I am attaching a pdf with some additional suggestions.

We would like to thank the referee for taking the time to review our manuscript and for providing a commented version.

In the revised manuscript, we provide information regarding the students' evaluation, in lines 81-85. In Results section, we have also added a new figure (Fig. 7), with some representative pie charts on the students' responses.

Additionally, we have tried to shorten the description of sites. We now provide less details and the justification of why each site was included. Some further details were also added on the design of the virtual fieldtrip (lines 75-80).

We also address the pdf comments below.

Line 19 You mention it at the end but perhaps adding here in your introduction that it provides the opportunity for those with disabilities to participate in STEM. I think it's a good point.

We have added a relevant statement in line 38 of the revised manuscript.

Line 68 I may have missed this but can you clarify who studied the bibliographical references? Was it the field trip developers or students?

It is now clarified in the text (line 68).

Line 69 This might just be personal preference but I prefer the term "sites" instead of "stops"

We have changed stops to sites throughout the text.

Line 76-77 This sentence is a bit awkward. You might consider rephrasing it to "134 third year students from the Faculty of Geology & Geo-environment participated in the virtual field trip."

The correction has been made, see line 81.

Line 76 Was it just presented or did the students engage with it? I'm assuming it is the latter but I would clarify.

It has been clarified in lines 81-82.

Line 79 You might consider entitling this "field trip focus and design" or something along those lines. The study area itself is not as important as what you created and how the students interacted with what you created. One - two basic sentences on the actual study area should be sufficient.

The title has been modified. The sites' description has been shortened.

Line 224-225 I'm assuming that this is self-report. I would say that unless you actually tested them.

The percentage corresponds to self-report. We have specified in the text, see line 205.

Line 228 Is this of the students who mentioned restrictions?

It is now specified in the text, see lines 197-198

Line 233-235 I'm not clear on what you are trying to say here. You might consider rephrasing these two sentences.

We have modified these sentences, see lines 202-204.

Lines 239-242 Did you assess this? If so, can you report on the outcomes?

Students were assessed through a post-field trip exercise and the final semester exams of the course. We provide details in lines 205-209.

Lines 248-251 is this from the feedback forms? Is there a demonstrative quote from any of the students that you can feature?

We have added a characteristic response from one of our students, in lines 224-226.

Line 252 Instead of other educational methods, I would suggest saying "traditional passive learning methods." There are many active learning strategies other than virtual field trips.

We have modified the sentence accordingly.

**Editor comments**

The review comments are generally positive and constructive, providing the authors some guidance for improving the paper. Both reviewers pointed out that the authors should provide more details on the construction and function of the VFT, with less emphasis on the geomorphology of each site.

The author's responses have been receptive and demonstrate that the authors are likely to make the suggested changes to improve the paper. The authors have already provided a sample figure in response to reviewer 1.

For this reason I have decided that this paper can be published in this special issue subject to receipt of a revised manuscript addressing the reviewers' comments.

In addition, I would suggest that the authors provide a legend entry for the dark grey category in the upper right hand pie chart in their new figure. I would anticipate that the text and/or figure caption provide a description of the question that resulted in this particular pie chart. For example, was it a free-form or did the students select from a list of options?

Thank you for the comments and the positive response. Regarding the new figure, it has been added to the revised manuscript (Fig. 7), the legend entry was corrected, and we specify in the caption that the students had to select from a list of options.